# Antiradical and Antioxidant Activity and Stimulation of Pancreatic Lipase by Extracts Obtained from Saponin-Rich Raw Materials: Experimental and In Silico Study

**DOI:** 10.3390/ijms262110254

**Published:** 2025-10-22

**Authors:** Zbigniew Sroka, Karina Kapusta, Klaudia Suchańska, Wojciech Kołodziejczyk, Beata Żbikowska, Andrzej Gamian, Kacper Strzelczyk, Michał Gleńsk, Kamil Wojciechowski

**Affiliations:** 1Department of Pharmacognosy and Herbal Medicines, Wroclaw Medical University, Borowska 211a, 50-556 Wroclaw, Poland; beata.zbikowska@umw.edu.pl (B.Ż.); michal.glensk@umw.edu.pl (M.G.); 2Department of Chemistry and Physics, Tougaloo College, Tougaloo, MS 39174, USA; kkapusta@tougaloo.edu; 3Student Faculty of Pharmacy, Wrocław Medical University, Borowska 211, Wrocław, Poland; klaudiassuchanska@gmail.com (K.S.);; 4Department of Chemistry, Physics and Atmospheric Sciences, Jackson State University, Jackson, MS 39217, USA; dziecial@icnanotox.org; 5Department of Immunology of Infectious Diseases, Hirszfeld Institute of Immunology and Experimental Therapy, Polish Academy of Sciences, Rudolfa Weigla 12, 53-114 Wrocław, Poland; andrzej.gamian@hirszfeld.pl; 6Department of Chemistry, University of Warmia and Mazury in Olsztyn, Pl. Łódzki 4, 10-721 Olsztyn, Poland; kamil.wojciechowski@pw.edu.pl

**Keywords:** pancreatin, lipase stimulation, plant extracts, saponins, antioxidants, plant phenols, computational study

## Abstract

The effects of saponin-rich extracts on pancreatic lipase activity were examined. Horse chestnut seed and ginseng root extracts stimulated lipase activity the most, while ivy leaf and fenugreek seed extracts had an inhibitory effect. The strongest antiradical activity (ABTS^+•^–2,2-azino-bis(3-ethylbenzothiazoline-6-sulfonic acid) test) was exhibited by horse chestnut seed extract and the weakest by the soapwort root extract. Antioxidant activity strongly positively correlated with the total amount of phenols. Experimental studies showed that the surface properties of saponins present in extracts influence emulsion stabilization and have a moderate stimulating or inhibitory effect on pancreatic lipase activity. Computational studies demonstrated that lipase activity is stimulated by stabilizing an oil emulsion in the case of all studied saponins. Depending on the type of saponin, activation or inhibition of lipase was observed due to the immediate ligand–enzyme interactions.

## 1. Introduction

Various digestive tract diseases are associated with impaired fat digestion [1,2,3]. One of the first stages of fat digestion takes place in the stomach, in the presence of gastric lipase [4,5], but the main step of lipid hydrolysis occurs in the duodenum [6] where digestive enzymes are secreted, including pancreatic lipase, which is the most important enzyme in the digestion of fats [7]. Fats hydrolysis takes place in the presence of natural surfactants such as sodium cholate, sodium taurocholate, deoxycholate, taurodeoxycholate, and chenodeoxycholate, but the major bile salt found in the largest amount in bile is sodium cholate [8]. Surfactants maintain the fat fraction as an emulsion causing highly developed surface contact between water and fat [8]. Surfactants with moderate activity have a stabilizing effect on the emulsion and increase fat hydrolysis [9]. Strong surfactants can inhibit lipase activity either by affecting the structure of the enzyme or by causing the breakdown of the lipid enzyme complex, which is important for the reaction to proceed [9].

Problems with fat digestion can be caused by various factors, such as impaired biliary tract function, insufficient bile production, or impaired function of the gallbladder or sphincter of Oddi, which are responsible for the penetration of the necessary amount of bile into the duodenum.

Impaired lipid digestion causes symptoms such as the feeling of fullness in the abdomen, bloating and reflux, which can be alleviated, among others, by supplementing pancreatin or bile salts.

Pancreatin is an extract from the pancreas containing the main digestive enzymes, including lipase [7,10]. Pancreatin supports the exocrine function of the pancreas and thus facilitates the digestion in the duodenum, including the digestion of fats [11].

Plants and plant extracts are sources rich in various chemical compounds with broad-spectrum biological and therapeutic activity [12]. An important group of compounds is saponins, which exhibit numerous therapeutic effects such as anti-inflammatory, sealing, and strengthening of blood vessels [13]. A common feature of saponins is strong surface tension-reducing activity [13], stabilizing the emulsion [14], thus increasing the contact of lipase with fats.

In addition to saponins, another important group of plant compounds is phenolics, including flavonoids. Due to their chemical structure, phenolics have strong antioxidant, antiradical, anti-inflammatory, antimicrobial, spasmolytic, and antihemorrhagic activity [15]. Thanks to their antiradical properties, they are believed to slow down the aging processes [16].

In this study, we selected nine raw materials rich in saponins that showed a significant effect on surface tension (foaming test). The extracts from these raw materials were prepared and their surface properties and effect on pancreatic lipase activity were examined.

Using in silico methods, we investigated the possible mechanism of action of selected saponins in the lipolysis process. The content of saponins in extracts we measured with the UHPLC method. Additionally, we examined the content of phenolic compounds and determined the antiradical and antioxidant activity of extracts.

## 2. Results

### 2.1. Influence of Extracts on Lipase Activity

The effect of extracts obtained from saponin-rich raw materials on the lipolytic activity of pancreatin is demonstrated in Table 1 as the values *RA*% (Appendix A) and *Effect*% (Figure 1) as well as in Figure 2a,b as the value C_FFA_ (values *RA*%, *Effect*% and C_FFA_ are explained in Section 4.5, Materials and Methods). In Table 1, the influence of sodium cholate on lipase activity is shown as the positive control.

During the study, it was observed that extracts rich in saponins both stimulate and inhibit the lipolytic activity of pancreatin. The highest stimulatory effect expressed as *RA*% was exhibited by the extract from horse chestnut seeds and the inhibitory effect for extract from common ivy leaves (Table 1 and Appendix A). Similarly, the highest lipase stimulating activity expressed as *Effect*% was observed for the extract from horse chestnut seeds, while extracts from common ivy leaves and fenugreek seeds inhibited lipase activity (negative values of *Effect*%), Table 1, Figure 1.

### 2.2. The Amount of General Phenolic Compounds and Flavonoids in Extracts

Amounts of general phenols and flavonoids were measured using the colorimetric methods described by Sari et al. [17]. The results are presented in Table 2 and Appendix A.

The largest amount of phenolic compounds was measured for R.a. extract obtained from rhizomes of butcher’s broom 243 ± 54.5 (GAE ± ME), while the lowest amount of phenolic compounds was found in the extract G.p. (39 ± 1.62, GAE ± ME).

Most flavonoids were identified in the extract G.g. (362 ± 1.323, QE ± ME), while the smallest amount of flavonoids was found in the extract G.p. (4.20 ± 0.25, QE ± ME).

### 2.3. Antioxidant and Antiradical Potential of Extracts

Plant raw materials contain phenolic compounds, in varying amounts, a significant number of which have strong antioxidant and antiradical properties [18]. We used two methods in our research, based on electron transfer (ET)—ABTS^+•^ (2,2 azino-bis(3-ethylbensothiazoline-6-sulfonic acid)) and FRAP (Ferric Reducing Antioxidant Power). Most of the tested extracts showed similar ABTS^+•^ radical scavenging abilities, except the extract from *Gypsophila paniculata* (common gypsophila), which quenched the radical the least (Table 2, Appendix A).

In the FRAP test, the extract R.a. exhibited the highest antiradical properties. Other extracts exhibited weaker antiradical activity, in the following decreasing order: G.g. > H.h. > P.g. > T.f.g. > Q.s. > A.h. > P.v. > G.p. (Table 2, Appendix A).

### 2.4. Equilibrium Surface Tension, Surface Elasticity Modulus, and Surface Viscosity Modulus

Since saponin-rich extracts are known to possess surfactant properties [19], the surface activity of the extracts employed in this study was determined to correlate with the observed effect on the lipase activity. Surface properties of extracts such as equilibrium surface tension, surface elasticity modulus, and surface viscosity modulus are presented in Appendix A. Because of the inherently interfacial nature of the lipolytic activity, involving mass transfers between the bulk oil and aqueous phases, and the interfacial layer, the majority of mechanistic studies reported so far have dealt with aqueous-oil interfaces. Although the latter much better reflects the real situation in the gastric system, it also limits the available range of attainable surface pressure values. In some of our measurements the equilibrium surface tension (γ_eq_) approached 20 mN/m (see below), which corresponds to surface pressure, Π_eq_ > 50 mN/m at the aqueous-air interface (Π_eq_ ≡ γ_0_ − γ_eq_, where the index “0” refers to the pure solvent (Milli-Q water), γ_0_ = 72.5 mN/m at 21 °C). Taking into account that the maximum attainable interfacial pressure for edible oils against water rarely exceeds 40 mN/m [20], the use of an aqueous oil system might seriously bias the surface activity measurements. To avoid experimental difficulties with very low interfacial tension and with oil purification, we deliberately chose the aqueous-air system for the present study. The equilibrium surface tension (γ_eq_) and surface compression storage (elastic, E′) and loss (viscous, E″) moduli of the extracts are presented in Appendix A. The extracts displayed different abilities to reduce surface tension, with γ_eq_ varying between ~30 and ~46 mN/m. The γ_eq_ value for sodium cholate (35.9 ± 0.3 mN/m) did not stand out from those of the extracts. A much higher variability could be observed for the surface compression rheology parameters, with the elastic modulus (E′) varying between 9.2 ± 0.2 mN/m (for G.p.) and 123.7 ± 0.6 mN/m (for R.a.). The lowest surface viscosity modulus (E″ = 2.8 ± 0.8 mN/m) was also observed for G.p. but the highest (E″ = 43.1 ± 4.0 mN/m) was observed for H.h., with R.a. showing the second highest E″ (15.0 ± 0.7 mN/m). As compared to the extracts, the positive control (sodium cholate) showed much lower surface compression rheology parameters (E′ = 12.7 ± 0.4 mN/m, E″ = 1.0 ± 0.3 mN/m).

As lipolysis is an interfacial process, the surface activity of pancreatin alone and in the presence of the extracts may shed some additional light on the mechanism of fat digestion supported by the saponin-rich extracts. To this aim, surface tension and surface compression rheology parameters were determined for pancreatin and its mixtures with the extracts analogously as described above (Appendix A). The enzyme mixture alone is only weakly soluble in water and its saturated solution reduced surface tension only to 50.9 mN/m. On the other hand, the adsorbed layer showed a significant elastic response (E′ = 47.8 mN/m). All mixtures of pancreatin with the extracts or cholate showed higher reduction in surface tension than the enzyme alone, and in some cases (R.a., G.g., G.p. and P.g.) even higher than for the extracts alone (Figure 3). As for the surface rheology parameters, in most cases E′ of the mixed pancreatin-extract layers remained close to that of the extract, except for Q.s., P.g., and T.f.g., where it exceeded the values of both the extract and pancreatin alone. Overall, the most pronounced synergistic effect on surface tension was observed for P.g., where γ_eq_ dropped by 12.4 mN/m with a very large increase in E′ (by 158.6 mN/m). At the other extreme, for H.h. the equilibrium surface tension increased by 11.4 mN/m when mixed with pancreatin, while the mixed layer showed E′ lower than for any of the mixture components alone. Among the investigated extracts, R.a. seems to be the least affected by the presence of pancreatin, as both γ_eq_ and E′ remained practically the same (and very high, above 120 mN/m).

### 2.5. Emulsion Stability in the Presence of Extracts

The effectiveness of the enzymatic lipolysis also depends on the efficacy of oil dispersion in the aqueous phase. To compare the emulsification capabilities of the investigated extracts, a series of emulsions was prepared using olive oil (stained with an oil-soluble red dye) under the same conditions. The results of the emulsification test are presented in Appendix A.

Immediately after homogenization, all samples looked homogeneous (pink, cream) indicating a transient formation of an emulsion. However, within 1 h, a clear, red-colored oil layer phase separated in all samples (Appendix A), indicating fast demulsification of excess oil in all samples. Within the next 24 h, the height of the phase-separated oil layer did not change significantly but the phase below it further separated into a water-rich lower phase and an oil-rich upper phase (Appendix A). The investigated extracts were thus capable of emulsifying different amounts of olive oil, although none of the emulsions could last for more than a few minutes. Estimating from the height of the phase-separated olive oil layer and the transparency of the bottom aqueous layer, the best emulsification was achieved for R.a., G.g. and A.h., and the worst for Q.s., P.v., and H.h. The positive control (sodium cholate) did not differ significantly from the extracts in its emulsifying capacity.

### 2.6. Qualitative Analysis of Compounds in the Most Active Extracts by UHPLC

The most active extracts were investigated for the content of phytochemical compounds, especially saponins known as surface-active compounds. These are extracts obtained from ginseng roots and chestnut seeds.

The most active extracts with a stimulatory effect on the lipolytic activity in the tested model in vitro were horse chestnut extract and ginseng extract. Therefore, these two extracts were analyzed by LC-MS in order to characterize and/or confirm their phytoconstituents. Eight distinctive peaks were observed in the LC-MS chromatogram of aqueous ethanol extract from the roots of *P. ginseng* and 20 distinctive peaks were observed in the LC-MS chromatogram of aqueous ethanol extract from the seeds of *A. hippocastanum*. The chromatograms are presented in Appendix A, whereas their tentative characterizations are shown in Appendix A.

### 2.7. In Silico Study of Emulsion Stabilization

A computational study was performed to investigate potential emulsion stabilization and lipase activation/inhibition by saponins. Several simplifications were considered to achieve the best accuracy-to-computational resources ratio for describing the in vitro system. Each plant extract was represented by only one major saponin constituent, such as hederacoside C for *Hedera helix* leaves [21] and escin Ia for *Aesculus hippocastanum* seeds [22]. The surfactant properties of these two compounds, along with the sodium cholate (a bile component) and blank (no saponins) systems, were assessed through 100 ns molecular dynamics (MD) simulations of the disordered oil-water system (Appendix A).

During the simulation, the blank system exhibited a distinct structure change in oil/water phases (Appendix A). Initially, an oil agglomerate formed as a quatrefoil or rhombic cross. After 40 ns of simulation, the oil phase transitioned into a cylindrical shape, which remained stable throughout the simulation with no significant movement from the center of the cell. Adding saponins or sodium cholate led to faster formation of a cylindrical-shaped oil phase (within ten nanoseconds on average), which moved flexibly within the simulation cell (leaving the cell from one side and entering from the other). Both sodium cholate and saponin molecules consistently positioned themselves at the water/oil interface.

The delayed formation and stability of the oil cylinder in the blank system suggests that, in the absence of emulsifying agents, the oil and water phases interacted minimally, leading to faster phase separation. Conversely, the oil cylinder’s faster formation and dynamic movement in the presence of saponins indicated their role in reducing interfacial tension and enhancing emulsion stability. The addition of the sodium cholate (Appendix A) and escin Ia (Appendix A) showed similar behavior in promoting oil cylinder formation and its movement, with escin Ia having a slightly weaker effect. Some difference was observed for hederacoside C (Appendix A). Its simulation showed the rapid formation of smaller oil agglomerates, which rapidly merged with the main oil cylinder. The frequent separation followed by rapid emergence could indicate that the emulsion lacked stability, indicating less established emulsification properties.

### 2.8. The Effect of Saponins on the Lipolytic Activity of Pancreatic Lipase Through Computational Studies

To further examine the differences in saponin activities, their mechanism of interactions with the porcine lipase–colipase was investigated. The crystallographic structure with PDB ID: 1ETH was retrieved from the Protein Data Bank. It comprised lipase, colipase, tetraethylene glycol monooctyl ether (TGME) bound to the active site of the lipase, and two oligosaccharide molecules (beta-D-mannopyranose-(1-3)-[beta-D-mannopyranose-(1-6)]beta-D-mannopyranose-(1-4)-2-acetamido-2-deoxy-beta-D-glucopyranose-(1-4)-2-acetamido-2-deoxy-beta-D-glucopyranose) at the glycosylation site (Figure 4a). As discovered by Hermoso et al. [23], the nonionic detergent TGME, besides its surfactant properties, may have an inhibitory effect on pancreatic lipase by binding to its active site at sub-micellar concentrations. The linear shape of this molecule allows it to enter the enzyme’s active site. The saponins studied here, meanwhile, were too large to enter the binding pocket of an active site. Since saponins are glycosidic compounds [24], we used the glycosylation sites to dock three new ligands to the lipase–colipase complex. One of the sites was near the N-terminal, while the other was near the active site just across the lipase loop β-9 (Figure 4b). Four systems were further subjected to a molecular dynamics simulation: lipase–colipase (chain A and chain B), lipase–colipase (LC)-cholate, LC–hederacoside C, and LC–escin Ia (consisting of chain A and chain B and two of each of the following: cholate, hederacoside C, and escin Ia, respectively). As a result of the 100 ns simulation, the LC–hederacoside C complex showed the smallest movement and minimal root mean square deviations (RMSDs) (Figure 4c). Lipase–colipase exhibited relatively high deviations from the beginning of the simulation, with significant fluctuations. Nonetheless, the RMSD fluctuation range did not exceed 2 Å, which can still be considered stable. Notably, this LC complex was modeled without its co-crystallized glycans, likely contributing to the reduced instability. Lipase–colipase complexes with escin Ia also exhibited high RMSD fluctuations, though it was stabilized after 80 ns of simulation time. The cholate system underwent a major conformational change at around 40 ns and reached stabilization only after 80 ns of simulation. Lower deviations of the LC–hederacoside complex suggest that its conformation remains close to the crystallographic reference structure of lipase–colipase in the inhibited state. In other words, binding of hederacoside does not induce structural rearrangements in the protein, which is consistent with its role in maintaining an inactive conformation. By contrast, the addition of escin Ia or cholate presumably led to pronounced structural changes, reflected in higher RMSD values and major conformational shifts, with stabilization occurring only after 80 ns. While a longer simulation time could be beneficial here, our primary objective was to capture whether ligand binding preserves or perturbs the inhibited conformation. The clear distinction between hederacoside (no conformational change, stable inhibition) and escin/cholate (conformational shift, possibly, toward an active-like state) provides sufficient evidence to support our hypothesis within the simulated timescale.

The most significant fluctuations for lipase–colipase and LC–escin Ia were observed in the area of the β-9-loop, while the lid was the most flexible area of LC-cholate (Figure 4d). No significant fluctuations were observed for the β-5-loop or a catalytic triad (RMSF < 1.3 Å) for all models.

Cholate bound near the N-terminal (referred to as 1st) showed the highest RMSD jump for a ligand fit on protein from its reference structure (Figure 4e), illustrating the possible relocation of the ligand with further stabilization. Meanwhile, the cholate near the active site (referred to as 2nd) remained with no significant changes, tightly kept just across the β-9-loop from the catalytic triad. LC–hederacoside C showed an opposite trend: while its 1st molecule was set tightly within the binding pocket of the glycans, the 2nd molecule moved to the enzyme’s active site after 50 ns simulation. It remained there, blocking the active site. As for escin Ia, its first molecule showed a somewhat stable RMSD profile, while the second escin Ia exhibited a stepwise increase in RMSD, indicating the relocation of the ligand away from the glycan position across the β-9-loop without any further stabilization. Except for the second escin Ia, all other ligands seemed to have relatively low RMSD after stabilization, indicating some stability of the obtained complexes.

For each ligand, 2D interaction diagrams for the simulation of the last 50 ns were analyzed (Figure 5). The 1st cholate molecule migrated from the original glycosylation side at the N-terminal towards residues ARG 66 and ARG 69, accepting hydrogen bonds (HBs) from these residues throughout 40–60% of the simulation time (Figure 5a). Located near the lipase’s active site, cholate did not show solid interactions either (Figure 5b), with the carboxylic group accepting HBs from LYS 233 (36–39% of the simulation time) and one of the hydroxyl groups donating HB to ALA 207 (only 35% of the time). The most substantial interactions were exhibited by hederacoside C within the glycosylation side at the N-terminal. One of the hydroxyl groups on its α-L-arabinopyranosyl ring formed hydrogen bonds (HBs) with CYS 10 for 90% of the simulation time and ARG 165 for 75%. Another hydroxyl group from the same ring formed two HBs with ARG 165, accounting for 170% (indicating the formation of more than one HB) of the simulation time, and a further hydrogen bond with LEU 8 for 97% of the time. The hydroxyl groups of the 6-deoxy-α-L-mannopyranosyl unit interacted through HBs with ARG 123 (39% and 49% for two separate groups) and ARG 164 (53%). Additionally, one of the hydroxyl groups was bound to GLU 161 and LEU 189 through water-mediated bridges. Throughout 42% of the simulation time, the ring oxygen maintained an H-bond with ARG 164, which also formed an additional HB with a hydroxyl group from the olean-12-en-28-oic acid backbone of hederacoside C. ARG 165 played a crucial role in bridging the α-L-arabinopyranosyl ring and the 6-deoxy-α-L-mannopyranosyl unit by forming interactions with both. At the same time, LEU 8 served as a connector to the β-D-glucopyranosyl unit. Moreover, one of the β-D-glucopyranosyl unit’s hydroxyl groups also formed an HB with ARG 7 for 88% of the time. These interactions collectively indicated strong and stable binding that kept hederacoside C bound to lipase. The second molecule of hederacoside C showed less significant interactions after it crossed the β-9 loop towards the active site of lipase (Figure 5d). The hederacoside C was kept near the active side according to the qualitative analysis of the trajectory, but only by H-bonding and water bridges between one of the ligand’s hydroxyl groups and lipase’s CYS 182, GLN 22, and GLN 184 throughout 32–39% of the simulation time (Figure 5c). The carboxylic group of the escin Ia near the N-terminal formed two hydrogen bonds with residues GLU188 and LEU189 within 97% and 83% of the simulation time, respectively (Figure 5e). The 2nd escin Ia did not interact with the protein, remaining fully exposed to the solvent throughout the simulation (Figure 5f).

The blockage of the active side solely by hederacoside C could suggest its inhibitory activity caused by direct blockade. On the other hand, the strong binding of this saponin near the N-terminal could have an allosteric effect on the activity of the lipase. To investigate it further, we measured the distances between the critical loops near the active side throughout the 100 ns MD trajectory, as shown in Figure 6. For comparison, one more model of a lipase without co-lipase was simulated. These measurements allowed us to evaluate the size of the substrate’s entryway or the opening of the lid.

The plot in Figure 6a illustrates the time progression of the triangle’s height (h), measured from the tip of the lid (Cα of ILE 252) towards the side defined as PHE 78-PHE 216 Cα’s distance between the loops β-5 and β-9 (Figure 6c). This measurement illustrates the extent of the lid. The largest height throughout the simulation time was noted in the LC-cholate model, with an average distance of 21.32 Å, exceeding the lipase–colipase model, which showed an average distance of 21.08 Å. The LC-cholate model also showed the highest fluctuation among the models, with a difference between the minimal and maximal height of 5.32 Å. The smallest opening of the lid was noted for the lipase without a colipase (average of 19.88 Å). The lid opening in the case of hederacoside C was larger than in the pure lipase but not as significant as in the case of the LC-cholate or lipase–colipase (average of 20.70 Å). Interestingly, even though the escin Ia showed minimal binding, the lid opening was reduced compared to the other models. The qualitative analyses of its trajectory revealed that, even though the lid did not open in the case of the LC–escin Ia model, the β-9 loop moved significantly (as shown in the RMSF plot in Figure 4d).

To better describe the entryway size, we also calculated the area of this triangle (Figure 6b) in addition to the previous method of calculating the triangle’s height, which only illustrates the lid extent. These measurements revealed the significant stepwise increase in the entryway due to the opening of the β-9 loop for the lipase–colipase complex in the presence of escin Ia. As in the previous case, the systems of pure lipase and the lipase–colipase complex in the presence of hederacoside C showed the smallest area of this triangle.

Finally, we measured the distances between the active site residues S154, D177, and H264 (Figure 6d). While the distance between S154 and H264 fluctuated significantly across all models, leading to some inconclusiveness, the distance between S154 and D177 was slightly higher for LC-cholate and LC–escin Ia compared to LC–hederacoside and even the lipase–colipase complex. Moreover, the D177–H264 distance analysis showed that, in the cases of LC-cholate and LC–escin Ia, this distance was significantly reduced to 3–4 Å and exhibited the least fluctuation. In contrast, for all other complexes, the distance remained around 6–7 Å. In the LC–escin Ia complex, residues separated after 20 ns but re-approached after 50 ns of the simulation, which correlated with a marked increase in the triangle area between the loops (also observed after 50 ns), as well as with the relocation of the ligand away from the glycan position across the β-9 loop (Figure 4e). These differences in active site residue distances may be indicators of lipase activation by cholate and escin Ia.

## 3. Discussion

Lipases are enzymes that digest water-insoluble substrates, fats [25]. For the digestion to be effective, the substrate (soluble in water) must be in the form of an emulsion in which the contact surface of the fat with the aqueous environment in which lipase is dissolved is largely developed [9]. Compounds that promote emulsion stability are surfactants [26].

Strong surfactants, such as cetylpyridinium chloride, may inhibit lipase activity [9]. Slightly weaker surfactants, which include bile salts, have a positive effect on the enzymatic lipolysis of fats [27]. There is a group of chemical compounds in plants with a relatively similar structure to bile acids, which are known for their surface activity. These compounds are saponins [28]. The majority of the saponin-rich extracts employed in this study showed significant surface activity, with γ_eq_ values in the range overlapping with that of the duodenal (27.8–35.4 mN/m) and jejunum fluids (~30 mN/m) [29]. The γ_eq_ values for many extracts also approached those observed for the popular synthetic low-molecular weight surfactants, such as anionic sodium dodecylsulfate (SDS), cationic cetyl trimethylammonium chloride (CTAB) and non-ionic ethoxylated alcohol surfactants [30,31], and were much lower than for pancreatin alone. The equilibrium surface tension of 1% sodium cholate, employed as a positive control in the present study, also showed a similar value, 35.9 mN/m, which agrees well with the data reported previously [32].

Our previous experiments showed that extracts from plant raw materials rich in saponins stimulated the activity of pancreatic lipase [33]. These extracts also had beneficial effects in the presence of sodium cholate. Moreover, the effects of the extracts and sodium cholate were additive [33]. The literature data are inconsistent; some indicate the inhibition of lipase activity by certain saponins [34], whereas other studies show a clearly beneficial effect of saponins on the activity of lipases [33].

We created a model system that reflects the duodenum conditions with the ability to control the content of the enzyme (pancreatin) and sodium cholate (the main component of bile). We investigated the effect of plant extracts rich in saponins on the intensity of fat digestion at a constant concentration of pancreatin (1.36 mg/mL) and in the absence of sodium cholate.

The aim of the research was to determine the extent to which an extract rich in saponins can replace sodium cholate, i.e., improve fat digestion in the absence of cholate.

In our research, we created a model system reflecting the conditions in the duodenum with the ability to control the content of the enzyme (pancreatin) and sodium cholate, which is the main component of bile.

Although the two most effective extracts (P.g. and A.h.) displayed the lowest equilibrium surface tension among the tested extracts (<32 mN/m), the lipase-supporting activity is apparently not directly correlated with γ_eq_. For example, P.v.—one of the least effective extracts—showed γ_eq_ similar to the control sodium cholate, whose enzyme-supporting effect was two orders of magnitude higher than for P.v.

Besides the ability to reduce surface tension, saponins and saponin-rich extracts are also known for formation of highly viscoelastic adsorbed layers, often featuring high surface elasticity and viscosity compression moduli (E′, E″) [35,36]. However, the surface rheology parameters are not the key factors in determining the effect on lipolytic activity, either. Both E′ and E″ for the positive control were negligible as compared to most of the investigated extracts, especially R.a. and H.h. (both showing little or even a negative effect on pancreatin’s enzymatic activity). The E′ and E″ for sodium cholate were in fact closer to those of the low molecular weight synthetic surfactants, which are not able to form any viscoelastic adsorbed layers. The high surface viscosity modulus values (as observed, e.g., for H.h. or T.f.g.) seem even to inhibit pancreatin’s lipolytic activity.

Having established that surface activity of the extracts on its own cannot explain their effect on enzymatic fat hydrolysis, we looked for a correlation between the latter and the surface activity in the mixed extract–pancreatin systems. For the positive control (cholate-pancreatin mixture), the equilibrium surface tension and surface compression parameters were very—(Figure 3, Appendix A), leading us to hypothesize that the saponin-dominated interfacial layers might analogously favor pancreatin’s enzymatic activity. Such a situation can indeed be observed for R.a., although in this case the adsorbed layer showed much higher elasticity (E′ = 133.8 mN/m vs. 9.5 mN/m for cholate-pancreatin). According to Mekkaoui et al. [37], such high surface elasticity at elevated bulk concentration is an indication of the tight packing and hindered diffusional exchange between the adsorbed layer and the bulk [37]. Consequently, even though R.a. indeed enhanced pancreatin’s enzymatic activity (50%), the increase was much weaker than that observed for cholate, possibly because of the hindered exchange of the lipase and its substrates and products between the interface and the bulk. Similarly, the large rise in E′ for P.g. observed in the presence of pancreatin (from 22.1 mN/m to 180.7 mN/m) might be responsible for hindering the mass exchange with the interface, limiting the enzymatic activity enhancement to only 108%. It is difficult, however, to explain the outstanding performance of A.h. in supporting the lipolytic activity (240%) in terms of the surface activity. Even more surprising is the fact that the surface behavior of the H.h.-pancreatin mixture is very similar to that of A.h.-pancreatin, with the former showing the highest inhibitory effect of −50%. Overall, the surface activity in the mixed system does seem to play some role in determining pancreatin’s enzymatic activity, although this effect cannot be fully reproduced in our present model system. One possible reason is the absence of the oil phase in our model, but one should not forget that the chemical interactions between the components of the mixture might also affect the enzymatic turnover.

The observed differences in lipase-supporting activity of the saponin-rich extracts could not be explained based on the formation of emulsions, either. Even though all emulsions were prepared under the same conditions, the emulsification capacity of the extracts did not correlate with either the surface tension or surface compression rheology parameters or with their effect on the pancreatic lipase activity. This highlights the complexity of the interfacial lipase activity in the presence of surface-active plant extracts enriched in saponins. This poor correlation might suggest that the effect of the saponin-rich extracts on pancreatin should be considered not only from a simple physicochemical point of view. For example, surface tension, surface elasticity, and viscosity moduli, as well as emulsification capacity for sodium cholate, were comparable to those of the extracts (often even inferior), yet the former performed much better in supporting the lipolytic activity of pancreatin. Therefore, even if the interfacial characteristics of the saponin-rich extracts were favorable, their chemical interaction with the enzyme might inhibit its activity.

The computational studies performed here proved that lipase activity is stimulated by stabilizing an oil emulsion in the case of all studied saponins. In silico investigation of the emulsion stabilization revealed the differences in the dynamics of the oil-water systems with and without adding saponins (or bile). In the case of the blank system, water molecules showed significant movement. At the same time, the oil phase remained in the center of the cell, suggesting fewer water-oil interactions and a lower chance for dispersion. When sodium cholate or two studied saponins were added, the oil phase moved along with water molecules, suggesting more interfacial interactions and a greater chance of dispersion. Interestingly, while the dynamics of sodium cholate and escin Ia systems had similar trends, slightly different behavior was noted for the hederacoside C system. This qualitative MD analysis correlated well with the experimental emulsification tests, which showed *Hedera helix folium* (hederacoside C in this simulation) as inferior to *Aesculus hippocastanum semen* (escin Ia in this simulation) and bile in achieving emulsification. Nonetheless, this type of analysis did not clarify the reasons for the complete inactivation of pancreatic lipase by *Hedera helix* folium extract or the differences in action between sodium cholate and *Aesculus hippocastanum* semen extract.

It was also found that the saponins’ action was not limited solely to their emulsification properties but also involved their direct interactions with the enzyme. According to the modeling studies, sodium cholate and escin Ia could bind the lipase with moderate strength, resulting in the substrate’s entry lid opening (or β-9 loop opening) and a noticeable reduction in the distance between active site residues D177 and H264. In contrast, hederacoside C showed the ability to bind both at the enzyme’s active site and at its glycosylation side near the N-terminus, exhibiting strong and stable interactions. Our modeling, therefore, suggests that sodium cholate and escin Ia may act as allosteric activators, whereas hederacoside C may act as an active site inhibitor. Nonetheless, the molecular dynamics simulations performed here may not have been long enough to capture all the nuances of these interactions. Despite of our simplification that the plant extract was represented by its major saponin, this modeling worked surprisingly well in predicting activation/inhibition of the porcine lipase–colipase complex, correlating well with experimental activity findings, which showed the highest activation by sodium cholate, moderate activation by *Aesculus hippocastanum semen* (escin Ia as a dominating saponin) and inhibition of the enzyme by *Hedera helix folium* extract (hederacoside C as a dominant saponin). These observations provide further insight into the mechanisms of lipase modulation, suggesting that inhibition may result from direct obstruction of the active site, potentially coupled with allosteric effects when a saponin binds to a glycosylation site near the N-terminus. In contrast, the activation of lipase activity by saponins appears to require ligands that lack the ability to bind at the active site.

Free radical reactions and oxidative processes play a significant role in the pathogenesis of many serious diseases, including cancer. Phenols are the main natural compounds responsible for antioxidant and antiradical processes in vivo. In our study we examined the content of total phenols and antioxidant properties of the extracts. The study revealed a positive correlation (r^2^ = 0.7577) between the content of total phenols (GAE) and the antioxidant properties (Fe^2+^ [mg/mL]) of extracts. No correlation was observed between the antioxidant properties of the extracts and the flavonoid content (r^2^ = 0.0035, Appendix A).

## 4. Materials and Methods

### 4.1. Raw Materials

The raw materials were selected based on their significant surface activity, previously determined by a foaming test (unpublished results).
**Botanical Name****of Species****English Common Name****Plant Part****English (*Latin*)****Abbreviation for Extract Used Herein****Weight of Extract [g]***Ruscus aculeatus*Butcher’s broomRhizome (*rhizoma*)R.a.2.05*Quillaja saponaria*Soap bark treeBark (*cortex*)Q.s.1.60*Gypsophila paniculata*Common gypsophilaRoot (*radix*)G.p.1.99*Panax ginseng*GinsengRoot (*radix*)P.g.2.05*Glycyrrhiza glabra*LicoriceRoot (*radix*)G.g.1.87*Primula veris*Cowslip (Oxlip)Root (*radix*)P.v.1.97*Hedera helix*IvyLeaf (*folium*)H.h.1.87*Aesculus hippocastanum*Horse chestnutSeed (*semen*)A.h.1.72*Trigonella foenum-graecum*FenugreekSeed (*semen*)T.f.g.1.85

### 4.2. Reagents

*Pancreatin* was bought in Sigma Aldrich, Merck, Germany, with the following ingredients: pancreatic lipase 40 PhEur unit/mg; pancreatic amylase 106 PhEur unit/mg; and pancreatic proteases (trypsin, chymotrypsin, elastase) 7.04 PhEur unit/mg.

Tris (hydroxymethyl)aminomethane—Roche, Germany, HCl, NaOH—Chempur, Gliwice, Poland, olive oil—goccia d’oro Extra virgin olive oil (F.lli Ruata S.p.A. Frazione Baroli, EXP 01.12.2023, l. 000608-4, Baldissero d’Alba, Italy), rectified ethanol 95%, Polmos Companies of the Spirytus Industry in Warsaw, Poland, thymolphtalein Merck, Darmstadt, Germany.

### 4.3. Apparatus

Vacuum evaporator Buchi (Buchi, Rotavapor R-100, Flawil, Switzerland);Laboratory scales (Radwag, AS 82/220, Radom, Poland);Lyophilizer (Christ Alpha 1-2 LO Plus, Frankfurt, Germany);Water bath with precise temperature control, (AJL, LW 102, Kraków, Poland);pH meter (Oakton, WD-35419-03, Vernon Hills, IL, USA);Laboratory water purification unit Integral 10 Milli-Q (Millipore, Burlington, VT, USA);Profile Analysis Tensiometer PAT-1 (Sinterface Technologies, Berlin, Germany);Ultrasonic probe Sonopuls HD 2070 (Bandelin, Berlin, Germany).

### 4.4. Preparation of Extracts

10 g of dry raw material was extracted with 80 mL 70% ethanol solution in water for 10 days using a reflux condenser at 40 °C. Extracts were concentrated to dryness under reduced pressure and lyophilized.

Weight of extracts and abbreviations denoting extracts used in the manuscript are given in the Section 4.1.

### 4.5. Test of Activity of Pancreatic Lipase

Lipase activity was measured by the strongly modified test described by Tietz and Fiereck [38]. Eight-milliliter portions of extracts in 0.04 M Tris-HCl buffer, pH 8.0 at the concentrations 0.84, 1.69, and 3.38 mg/mL were added to Erlenmeyer flasks. In the control sample, 8 mL of buffer (without extract) was added to the flask instead of the extract solution. Then 2 mL of olive oil was added to each sample. All samples were shaken until the emulsion was formed and placed in a 37 °C water bath for 10 min of preincubation. To start the reaction, 1 mL of pancreatin solution in 0.04 Tris HCl buffer, pH 8.0, at 15 mg/mL was added to each flask. After 180 min of incubation at 37 °C, 3 mL of 95% ethanol in water was added to each flask to stop the enzymatic reaction. The level of fatty acids released by lipase was determined by titration of all samples with 0.05 M NaOH against 1% thymolphthalein in ethanol. Blank tests were also performed in which the enzyme solution was added to the samples after the addition of ethanol immediately before the titration of the samples. The reaction rate was expressed as the number of μmoles of fatty acids released by lipase calculated per mg of pancreatin using Formula (1).(1)$$C_{FFA}=3.333\cdot V_{ml}$$
where

C_FFA_ is the number of μmoles of fatty acids released by lipase calculated per mg of pancreatin; V_ml_ is the volume of 0.05 N NaOH used for titrating fatty acids released by lipase in the sample.

The influence of extract on lipase activity was also shown as a percentage change in the activity relative to the control (without extract), calculated according to the Formula (2):(2)$$RA\%=\frac{C_{\mathrm{FFA}}}{C_{\mathrm{FFA}0}}\cdot 100\%$$
where *RA*% is the relative activity of the sample compared to the control test (without extract), C_FFA_ is the number of μmoles of released fatty acids per mg of pancreatin at the extract concentration 3.38 mg/mL, and C_FFA0_ is the number of μmoles of fatty acids released in the sample without extract.

We also calculated the value marked as *Effect*% according to Equation (3). This value shows the increase or decrease in enzyme activity in the presence of extract in relation to the control sample (without extract). This value is expressed as a percentage and has positive values when the enzyme is stimulated or negative when enzyme is inhibited.(3)$$Effect\%=\frac{C_{\mathrm{FFA}}-C_{\mathrm{FFA}0}}{C_{\mathrm{FFA}0}}\cdot 100$$
where *Effect*% shows lipase stimulation or inhibition, C_FFA_ is the number of μmoles of released fatty acids per mg of pancreatin at the extract concentration 3.38 mg/mL, and C_FFA0_ is the number of μmoles of fatty acids released in the sample without extract.

### 4.6. Colorimetric Measurement of Total Phenolic Compounds

The amount of general phenols was measured with the method described by Sari et al. [17] with modification. A total of 200 μL of extract solution in 70% methanol at 20 mg/mL was added to the Eppendorf test tube, then 40 μL of Folin–Ciocalteu phenol reagent and 800 μL of 10% Na_2_CO_3_ in water were successively added. Samples were incubated at room temperature for 30 min and refrigerated after centrifugation at 12600 RCF. Then, 50 μL of each sample was added to the wells of the microwell plate and the absorbance was measured at 725 nm using a microplate reader. Blank samples with 70% methanol instead of extract solution were also prepared. The amount of phenols was shown as Gallic Acid Equivalents (GAE). All measurements were repeated three times.

### 4.7. Colorimetric Measurement of Flavonoids Using the AlCl_3_ Method

Measurement of flavonoids was performed with the modified method described by Sari et al. [17]. An amount of 50 μL of extract solution in 70% methanol at 20 mg/mL was added to the wells of the microwell plate. Then 50 μL of 2% AlCl_3_ in methanol was added to each sample and samples were stored at room temperature in the dark for 60 min. The absorbency was measured using a microplate reader at λ = 420 nm. The blank was prepared with 70% methanol instead of extract solution. The amount of flavonoids was shown as Quercetin Equivalents (QE). Measurements for each sample were repeated three times.

### 4.8. Measurements of Free Radical Scavenging Activity of Extracts Using the ABTS^+•^ Radical

The antiradical activity of extracts was measured by the method described by Le Grandois et al. [39]. A 7 mmol/L solution of ABTS in water and 2.45 mmol/L sodium persulfate in water were prepared. Then the solutions were mixed in a ratio of 1:1 and stored in the dark for 16 h to allow formation of the ABTS^+•^ cation radical. 200 μL of ABTS^+•^ radical solution and 2 μL of extract solution in 70% MeOH at 20 mg/mL were added to the wells in a microwell plate. After 15 min the absorbency was measured at λ = 734 nm. The measurements were repeated three times. The antiradical activity of extracts was expressed as Trolox equivalents, Tx [mM].

### 4.9. Antioxidant Activity of Extracts Measured Using the FRAP Method

The research was performed using the method described by Jimenez-Alvarez et al. [40]. To obtain FRAP reagent 78 mg of TPTZ (2,4,6-tris(2-pirydylo)-s-triazine and 132 mg of FeCl_3_ hexahydrate were dissolved in 25 mL of 0.3 M acetate buffer, pH 3.6 (CH_3_COOH-CH_3_COONa). The test sample was performed as follows: 20 μL of extracts dissolved in water (6.67 mg/mL) and 200 μL of FRAP reagent were added to the wells of the microwell plate. After 4 min, absorption was measured at λ = 593 nm. The results were presented as mg of Fe^2+^ in mL of sample (Fe^2+^ [mg/mL]), calculated based on the standard curve. All measurements was three time repeated.

### 4.10. Measurement of Equilibrium Surface Tension, Surface Elasticity Modulus, and Surface Viscosity Modulus

Milli-Q water was used to prepare all solutions for surface tension and rheology measurements. Its surface purity was checked by monitoring dynamic surface tension for 1 h. The extract powders alone or mixed with pancreatin were dissolved in Milli-Q water (1% each) and filtered through a 5 μm syringe filter immediately prior to the surface tension/surface rheology measurement using a PAT-1 drop profile analysis tensiometer. Temperature was maintained at 21 °C with a thermostatic bath. A drop of the tested extract (or extract+pancreatin) solution (10 µL) was formed at the tip of a steel capillary immersed in a glass cuvette (20 mL) filled with air. All experiments were performed at least four times (typically six times). In the first part of the measurement (0–600 s) the drop volume was kept constant (10 µL), providing information about the dynamic surface tension (i.e., surface tension vs. time). The equilibrium surface tension (γ_eq_) was calculated by extrapolating the dynamic surface tension to the infinite time following the approach of Joos and Hansen [41]. In the second part of each measurement, the drop area was subjected to sinusoidal oscillations with a relative amplitude of 4% and frequency of 0.1 Hz. The analysis of the surface tension response to the drop area oscillations provided the elastic (E′) and viscous (E″) parts of the complex surface compression viscoelastic modulus (surface elasticity modulus and surface viscosity modulus) [42].

### 4.11. Emulsification Tests

A sample of 1 mL of 1% extract solution of each plant extract or sodium cholate was homogenized with 0.2 mL of olive oil stained with Sudan Red IV (0.02%) using a Bandelin Sonopuls HD 2070 ultrasonic probe (30 s, 20% cycle, 30% max. power). Emulsions were stored at room temperature and photographed after 1 h and 24 h.

### 4.12. Analysis of the Amount of Saponin Compounds in Extracts Using UHPLC-ESI-MS and MS/MS

For ultra-high-performance liquid chromatography–electrospray ionization mass spectrometry (UHPLC-ESI-MS) analysis, samples were dissolved in 70% methanol (1 mg of the sample per 1 mL), filtered through a 0.22 µm PTFE syringe filter (Merck-Millipore, Darmstadt, Germany), and stored at room temperature before the analysis. Analytical UHPLC separation was conducted on the same day using the Thermo Scientific UHPLC Ultimate 3000 apparatus (Thermo Fisher Scientific, Waltham, MA, USA) consisting of an LPG-3400RS quaternary pump with a vacuum degasser, a WPS-3000RS autosampler, and a TCC-3000SD column oven. The system was linked to the ESI-qTOF Compact HRMS detector (Bruker Daltonics, Bremen, Germany). Samples were separated on a Kinetex RP-18 column (100 mm × 2.1 mm × 2.6 µm; Phenomenex, Torrance, CA, USA). The UHPLC-ESI-MS system was operated in the negative mode and sodium formate cluster ions were used for mass calibration. Data Analysis software version 5.3 (Bruker Daltonics, Bremen, Germany) was applied for data collection and evaluation of the obtained mass spectra. The main instrumental parameters were as follows: scan range 50–2200 *m*/*z*; dry gas—nitrogen; temperature 200 °C; potential between the spray needle and the orifice: 4.2 kV. Collision energy in CID cells was 35 eV. All chromatographic analyses were performed in a gradient mode. The solvent system consisted of solvents A (0.1% HCOOH in water) and B (0.1% HCOOH in acetonitrile). The injection volume was 2 µL and the flow rate was 0.3 mL/min. The following elution program was used: 0→30 min (5→95% B), 30→40 min (95% B), 40→45 min (95→5% B), and 45→50 min (5% B). UHPLC analyses were carried out isothermally at 30 °C.

### 4.13. Computational Details

All calculations were performed using the Schrödinger Software Package (Schrödinger Release 2024-1: Schrödinger, LLC, New York, NY, USA, 2024). Simplified disordered systems made of 20,000 molecules were generated to mimic the emulsions in four scenarios: (1) blank system—containing 92 molecules of triolein (to represent olive oil), 14 molecules of Tris, 8 H^+^ and 8 Cl^−^ ions (to represent Tris HCl) and 19,878 water molecules; (2) cholate system—containing 92 molecules of triolein, 14 molecules of Tris, 8 H^+^ and 8 Cl^−^ ions, 19870 water molecules, 4 Na^+^ and 4 cholate ions (to represent emulsion stabilized by sodium cholate); (3) hederacoside C system—containing 92 molecules of triolein, 14 molecules of Tris, 8 H^+^ and 8 Cl^−^ ions, 19,876 water molecules, and 2 hederacoside C molecules (to represent emulsion stabilized by hederacoside C), and escin Ia system—containing 92 molecules of triolein, 14 molecules of Tris, 8 H^+^ and 8 Cl^−^ ions, 19,876 water molecules, and 2 molecules of escin Ia. All structures were retrieved from the PubChem database (https://pubchem.ncbi.nlm.nih.gov/, accessed on 1 May 2024). Obtained models were used to perform a 100 ns Molecular Dynamics simulation using a Desmond Module [43]. The OPLS4e force field was utilized to perform the simulation under NPT conditions at a target temperature of 310.0 K (to mimic the internal body temperature) and pressure of 1.01325 bar, with a recording interval of 25 ps. Prior to the simulation, the model was relaxed using a default relaxation protocol. This protocol consisted of six steps. First, the system was minimized with solute restraints, followed by minimization without restraints. Next, an NVT ensemble with a Berendsen thermostat was applied for 12 ps at 10 K. This was followed by a 12 ps NPT ensemble using a Berendsen thermostat and barostat at 10 K and 1 atm. The fifth step involved a 24 ps NPT simulation at 300 K and 1 atm, still with the Berendsen thermostat and barostat. All ensembles included restraints on non-hydrogen solute atoms and velocity resampling every 1 ps. Finally, the system was subjected to an additional 24 ps of NPT simulation with a normal pressure relaxation constant and no restraints. We used the Maestro interface to analyze the trajectories obtained from the simulation and performed clustering based on the triolein molecules’ root mean square deviation (RMSD).

Molecular mechanics and molecular dynamics studies were carried out to investigate the potential influence of saponins on pancreatic lipase activity following the molecular docking studies. The 3D structure of the triacylglycerol lipase/colipase complex (PDB ID: 1ETH) was retrieved from the RCSB Protein Data Bank [44]. Repeated chains (C and D) were deleted from the tetrameric structure, as well as all water molecules and entries associated with the proteins’ chains C and D. Protein Preparation Wizard was used to ensure the reliability and accuracy of the protein structures. The tetraethylene glycol monooctyl ether (TGME) molecule was removed from the catalytic site. The original hydrogen atoms were replaced, bond orders were reassigned, and protonation states were generated at the target physiological pH = 7.4 with Epik [45]. Further, the hydrogen bond network was minimized, and the structures were optimized using the OPLS4 force field [46]. From this initial preparation, five systems were generated: (1) lipase–colipase—with two molecules of beta-D-mannopyranose-(1-3)-[beta-D-mannopyranose-(1-6)]beta-D-mannopyranose-(1-4)-2-acetamido-2-deoxy-beta-D-glucopyranose-(1-4)-2-acetamido-2-deoxy-beta-D-glucopyranose (chains E and F) being removed; (2) lipase–colipase–cholate (LC-cholate), (3) lipase–colipase–hederacoside C (LC–hederacoside C), and lipase–colipase–escin Ia (LC–escin Ia)—each with two molecules of beta-D-mannopyranose-(1-3)-[beta-D-mannopyranose-(1-6)]beta-D-mannopyranose-(1-4)-2-acetamido-2-deoxy-beta-D-glucopyranose-(1-4)-2-acetamido-2-deoxy-beta-D-glucopyranose (chains E and F) being substituted using molecular docking with cholate, hederacoside C, and escin Ia molecules, respectively. Structures of cholate, hederacoside C, and escin Ia were prepared using the LigPrep tool to optimize compounds. Molecular docking was performed in two steps. The positioning of the glucopyranose derivative was recorded to become the center of a grid, followed by the removal of these molecules. The first grid was prepared centered on a first glucopyranose derivative with a length of 36 Å and a size of the inner box of 10 Å. Ligand docking was carried out using the Glide module with Extra Precision (XP) [47] flexibly with the OPLS4e force field. Each obtained complex was used to generate a new grid and perform docking of a second molecule centered on the second glucopyranose derivative. Each built model was further subjected to the System Builder. An orthorhombic box of absolute size 120 × 80 × 80 Å was solvated with single-point charge (SPC) water molecules around the complexes. Systems were neutralized with counter-ions and subjected to the standard eight-step relaxation protocol followed by the 100 ns actual run with a 25 ps recording time step using the NPT ensemble class. We used the Simulation Interaction Diagram and Simulation Event Analysis tools to analyze the obtained MD trajectories. It should be noted that large glycosidic saponins such as escin Ia and hederacoside present particular challenges for molecular modeling. Although the OPLS4e force field has been parametrized for proteins, lipids, and many small molecules, its accuracy in describing highly flexible glycosides with multiple sugar moieties and stereocenters is more limited. This limitation may affect the precision of conformational sampling and interaction energies. Therefore, the results presented here should be interpreted comparatively, focusing on relative trends rather than absolute quantitative values.

System lipolytic activity was assessed by measuring the lid opening and entryway opening. To measure the extent of the lid opening, the triangle that connects the tips of loops β-5 and β-9 and the tip of the lid were chosen. The distances were measured between Cαs of the residues PHE 78, PHE 216, and ILE 252 (tips of the loops). The area of the triangle was calculated. The height of this triangle, which was pointed from the tip of the lid, was measured using Heron’s formula and the formula for the area of a triangle. The height (h, Å) and the area of the triangle (S, Å^2^) were plotted against the simulation time for all models.

### 4.14. Statistical Analysis

The maximal error (ME) was calculated with the total differential method for each measurement. The non-parametric statistical test (Kruskal–Wallis test by ranks, one-way ANOVA on rank) was performed to calculate the statistical significance of differences between the samples. Post hoc tests were performed to indicate which groups were significantly different from each other. Dunn’s test was performed with the Benjamini–Hochberg correction for multiple comparisons. The groups that differ significantly are marked with the letters a, b, c…, etc.

## 5. Conclusions

Two of the nine extracts studied showed a significant increase in pancreatin lipolytic activity.Even the most beneficial extract was almost four times less active than sodium cholate, which was used as a substitute for bile in our study.The above data show that even the most active extracts can only partially improve fat digestion in bile deficiency.Experimental and computational studies have shown that the stimulation of lipase activity by the tested substances is not only associated with stabilizing the emulsion.The allosteric interactions of active compounds with the enzyme are highly important, as is the potential active site inhibition caused by certain saponins.Antioxidant activity of extracts is positively correlated with the total content of phenolic compounds.

## Figures and Tables

**Figure 1 ijms-26-10254-f001:**
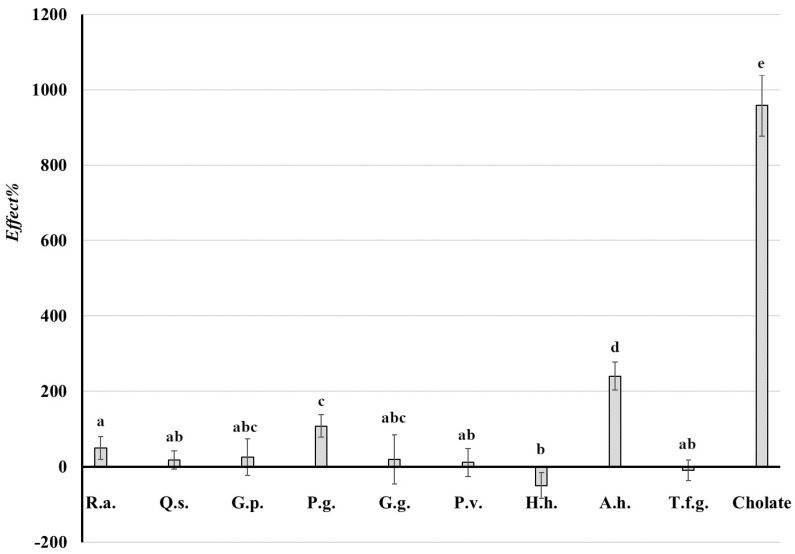
*Effect*% is the change in lipolysis concerning zero samples (without extract) and may become negative when lipase activity is inhibited or positive when enzyme activity is stimulated. R.a.—*Ruscus aculeatus*, Q.s.—*Quillaja saponaria*, G.p.—*Gypsophila paniculata,* P.g.—*Panax ginseng*, G.g.—*Glycyrrhiza glabra*, P.v.—*Primula veris,* H.h.—*Hedera helix*, A.h.—*Aesculus hippocastanum*, T.f.g.—*Trigonella foenum graecum.* There is no statistical significance of differences between bars marked with the same letter.

**Figure 2 ijms-26-10254-f002:**
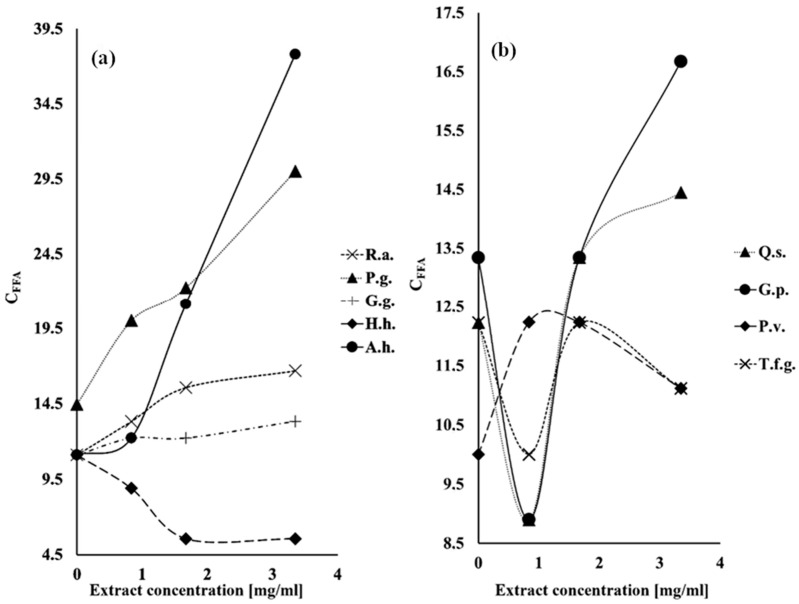
Effect of extracts obtained from selected saponin-rich raw materials on the lipolytic activity of pancreatin. The activity is expressed as C_FFA_ (the number of μmoles of fatty acids released by lipase calculated per mg of pancreatin), (**a**) R.a.—*Ruscus aculeatus*, P.g.—*Panax ginseng*, G.g.—*Glycyrrhiza glabra*, H.h.—*Hedera helix*, A.h.—*Aesculus hippocastanum*, (**b**) Q.s.—*Quillaja saponaria*, G.p.—*Gypsophila paniculata*, P.v.—*Primula veris,* T.f.g.—*Trigonella foenum graecum*.

**Figure 3 ijms-26-10254-f003:**
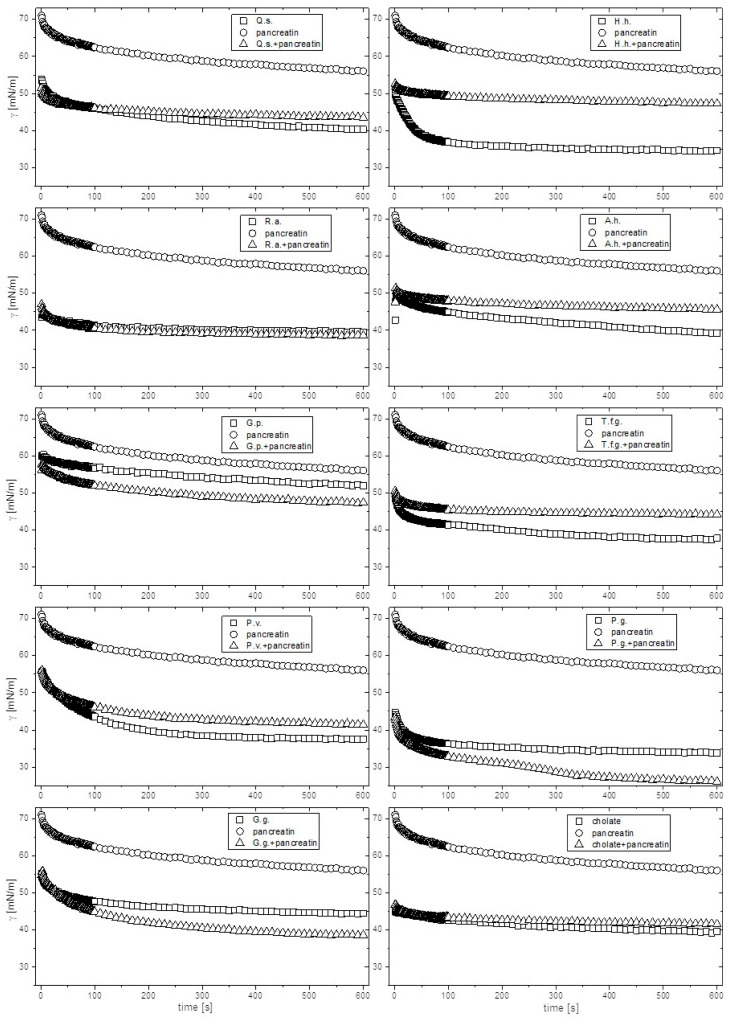
Dynamic surface tension for sodium cholate and the extract aqueous solutions and their mixtures with pancreatin (1% each). The curve for pancreatin alone is also shown for comparison. Q.s.—*Quillaja saponaria cortex*, R.a.—*Ruscus aculeatus rhizoma*, G.p.—*Gypsophila paniculata radix*, P.v.—*Primula veris radix*, G.g.—*Glycyrrhiza glabra radix*, H.h.—*Hedera helix*.

**Figure 4 ijms-26-10254-f004:**
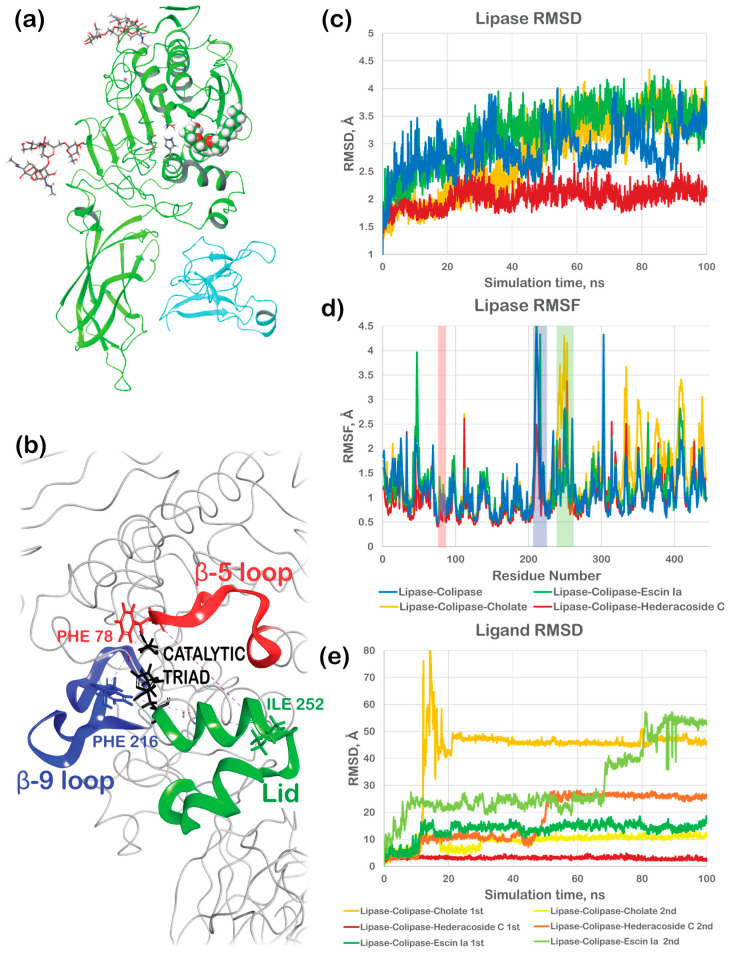
The results of 100 ns molecular dynamics simulation for five systems under investigation: (**a**)—reference lipase–colipase–TGME complex (PDB ID: 1ETH); (**b**)—zoom-in catalytic triad core and gateway loops β-5, β-9, and the lid (residues PHE 78, PHE 216 and ILE 252 are located at the tip of the loops and are used for distance measurements); (**c**)—root mean square deviations (RMSDs) of lipase throughout the simulation time; (**d**)—root mean square fluctuations (RMSFs) of lipase throughout the simulation time; (**e**)—root mean square deviations (RMSDs) of a ligand fit on lipase throughout the simulation time.

**Figure 5 ijms-26-10254-f005:**
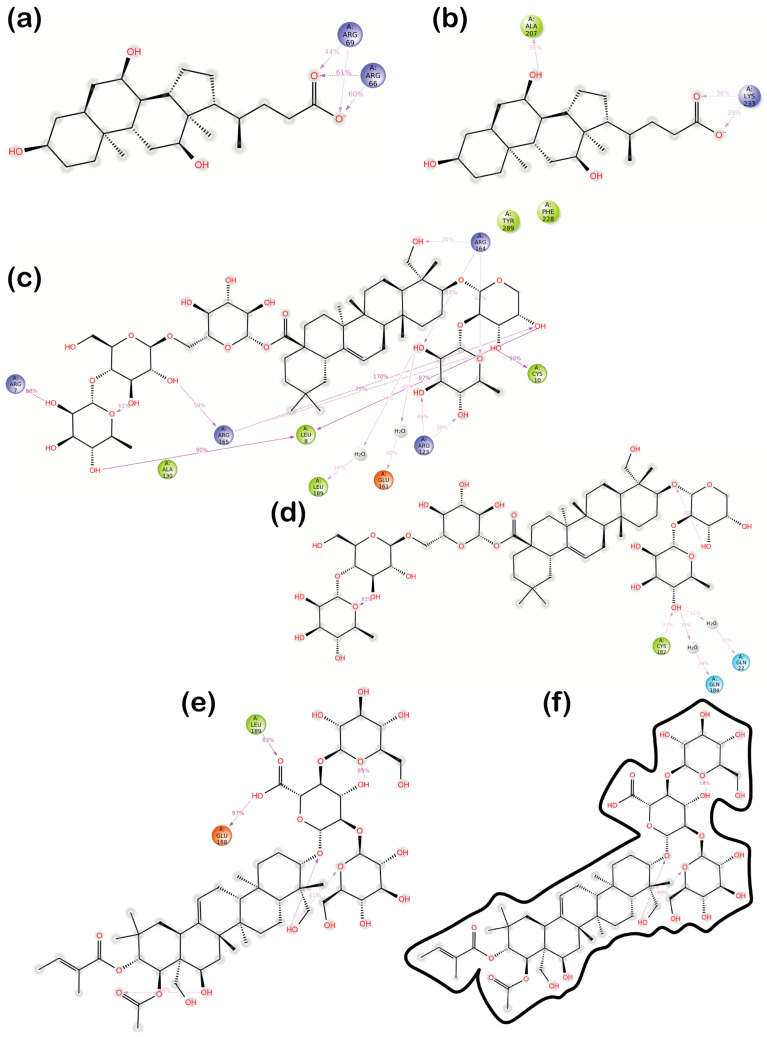
Two-dimensional interaction diagrams for the last 50 ns of the 100 ns MD simulation of LC-Cholate model: (**a**)—1st ligand and (**b**)—2nd ligand; LC–Hederacoside C model: (**c**)—2nd ligand and (**d**)—1st ligand; and LC–Escin Ia model: (**e**)—1st ligand and (**f**)—2nd ligand.

**Figure 6 ijms-26-10254-f006:**
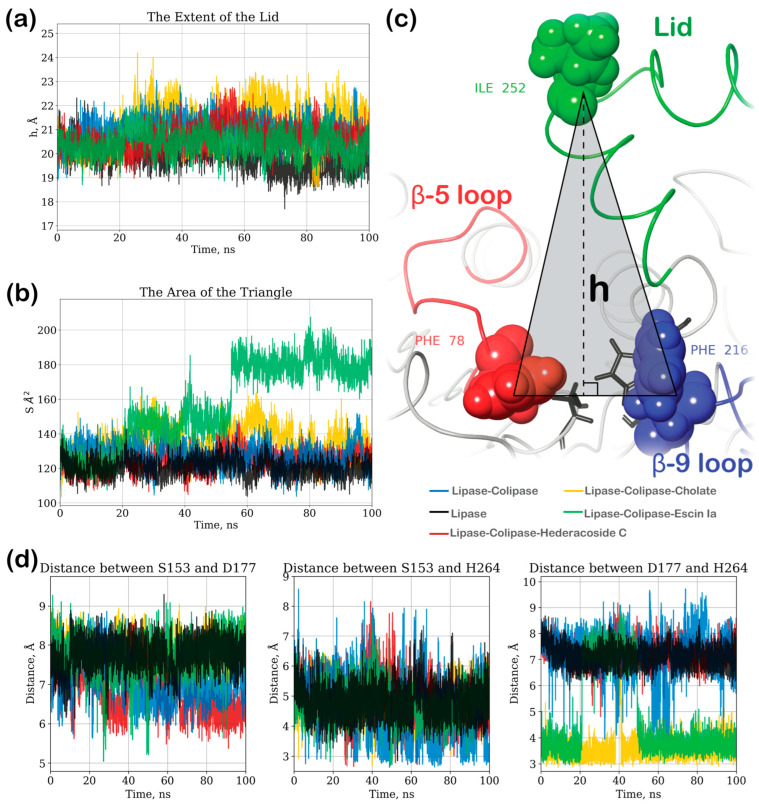
The analysis of the lipase’s active site measurements as a function of time for Lipase, Lipase–Colipase, LC–Cholate, LC–Hederacoside C, and LC–Escin Ia models: (**a**) the extent of the lid represented as a height (h) of the triangle between the tips of the gateway loops β-5, β-9, and the lid (measurement was made between Cα of PHE 78, PHE 216, and ILE 252, respectively) pointed from ILE 252; (**b**) the surface area of the triangle between the tips of the gateway loops β-5, β-9, and the lid; (**c**) illustration of the triangle used for measurement; (**d**) distances between active site residues.

**Table 1 ijms-26-10254-t001:** Effect of extracts on the lipolytic activity of pancreatin expressed as *RA*%, which is the relative activity of the sample compared to the control test (without extract); *Effect*% is the change in lipolysis concerning zero samples (without extract). It may become negative when lipase activity is inhibited or positive when enzyme activity is stimulated. ME is the maximal error, n = 3.

Extract	*RA*% ± ME	*Effect*% ± ME
**R.a.**	150 ± 30	50 ± 48
**Q.s.**	118 ± 24	18 ± 37
**G.p.**	125 ± 26	25 ± 63
**P.g.**	207 ± 29	107 ± 49
**G.g.**	120 ± 87	20 ± 78
**P.v.**	111 ± 41	11 ± 58
**H.h.**	50 ± 42	−49 ± 49
**A.h.**	340 ± 59	240 ± 82
**T.f.g.**	91 ± 38	−9 ± 46
**Cholate**	1058 ± 30	958 ± 81

**Table 2 ijms-26-10254-t002:** Amount of total phenols expressed as gallic acid equivalents (GAE), flavonoids as quercetin equivalents (QE), antiradical activity of extracts demonstrated as Trolox equivalents Tx [mM], reducing properties of extracts presented as ferrous ions (Fe^2+^) mg/mL. ME is the maximal error, n = 3.

Extract	GAE ± ME	QE ± ME	Tx [mM] ± ME	Fe^2+^ [mg/mL] ± ME
**R.a.**	243 ± 54.5	20.6 ± 4.24	1.37 ± 0.024	5.42 ± 0.192
**Q.s.**	116 ± 4	20.6 ± 1.32	1.28 ± 0.076	0.80 ± 0.057
**G.p.**	39 ± 1.62	4.20 ± 0.25	0.35 ± 0.061	0.19 ± 0.038
**P.g.**	95.4 ± 6.19	14.3 ± 0.715	1.36 ± 0.033	1.07 ± 0.052
**G.g.**	155 ± 22.0	362 ± 1	1.37 ± 0.025	2.03 ± 0.086
**P.v.**	142 ± 13.7	17.3 ± 2.65	1.27 ± 0.065	0.62 ± 0.048
**H.h.**	146 ± 6.20	90.0 ± 14.7	1.37 ± 0.026	1.34 ± 0.101
**A.h.**	131 ± 6.56	78.7 ± 6.57	1.37 ± 0.019	0.74 ± 0.066
**T.f.g.**	94.2 ± 4.43	164 ± 6.95	1.30 ± 0.071	0.82 ± 0.048

## Data Availability

Original contributions presented in this study are included in the text. Further inquiries can be directed to the corresponding author.

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
