# Peer review of "Antiradical and Antioxidant Activity and Stimulation of Pancreatic Lipase by Extracts Obtained from Saponin-Rich Raw Materials: Experimental and In Silico Study"

_ijms, 2025, doi:10.3390/ijms262110254_

Round 1

Reviewer 1 Report

Comments and Suggestions for Authors

The article is scientifically interesting. However, it lacks sufficient detail. I recommend publication after corrections.

-The authors cited references in the introduction up to 2024, including many recent articles. Nevertheless, they should also include more recent works (2025), as this is a topic of high scientific relevance.

-The image in Figure 5 has poor resolution, making it difficult to analyze the results. The authors should improve the image quality.

-Figure 4C shows the RMSD profile. In my analyses, the yellow and green lipase profiles only exhibited stability after 80 ns. The authors should provide a more thorough discussion of this behavior.

-In Figure 4C, the blue lipase profile indicates instability. The authors should discuss this aspect in greater detail in the manuscript.

-I suggest that the authors extend the molecular dynamics simulation time to provide greater clarity in the discussions. Alternatively, they may include more detailed discussions in the manuscript.

-Specific interactions such as hydrogen bonding and π–π interactions play a crucial role in molecular recognition. However, the authors did not provide an in-depth discussion of these interactions nor compare them with results from the literature. A more detailed discussion based on relevant studies should be included (e.g., 10.3390/molecules25122841 and 10.3390/molecules28196891).

Author Response

Dear Reviewer,

We greatly appreciate your time and effort in reviewing our manuscript and providing constructive feedback. Below, we provide detailed responses to each of your comments. We hope the revisions have improved the manuscript and made it suitable for publication.

Comments and Suggestions for Authors

The article is scientifically interesting. However, it lacks sufficient detail. I recommend publication after corrections.

-The authors cited references in the introduction up to 2024, including many recent articles. Nevertheless, they should also include more recent works (2025), as this is a topic of high scientific relevance.

A: The literature review included the latest work, which was selected as best as possible.

-The image in Figure 5 has poor resolution, making it difficult to analyze the results. The authors should improve the image quality.

A: The original figure was submitted to the editorial office in high resolution (300 px/in, size 7.2 × 9.8 cm). The reduced quality visible to the Reviewer most likely occurred when the image was saved within the DOCX file. We trust that the online version will display the figure in its original quality.

-Figure 4C shows the RMSD profile. In my analyses, the yellow and green lipase profiles only exhibited stability after 80 ns. The authors should provide a more thorough discussion of this behavior.

A: Indeed, the LC–escin Ia and LC–cholate complexes stabilized only after 80 ns. This behavior reflects ligand-induced rearrangements of the lipase-colipase complex, in contrast to hederacoside, which maintained the inhibited conformation. Following the Reviewer’s suggestion, we have discussed this in more detail in the Results section 2.8. to make it more straightforward for readers.

-In Figure 4C, the blue lipase profile indicates instability. The authors should discuss this aspect in greater detail in the manuscript.

A: Fluctuations in the lipase-colipase RMSD ranged between 2 and 4 Å (a 2 Å span), which can still be considered relatively stable. However, this appears less stable on the plot than the other systems, as the fluctuations are more pronounced. We attribute this behavior to removing the crystallographic glycans in this simulation, which may have reduced the overall stability. This point has been discussed in the Results section 2.8. per Reviewer’s suggestion.

-I suggest that the authors extend the molecular dynamics simulation time to provide greater clarity in the discussions. Alternatively, they may include more detailed discussions in the manuscript.

A: We agree that longer molecular dynamics simulations could provide additional insights; however, we could not extend them due to time constraints. Our primary objective was to determine whether ligand binding preserves or perturbs the inhibited conformation of the lipase-colipase complex. The simulations clearly showed that binding of hederacoside maintains the inhibited reference structure, whereas escin Ia and cholate induce conformational changes consistent with a shift away from inhibition. We have expanded the discussion in the manuscript (Results and Discussion sections) to clarify this point.

-Specific interactions such as hydrogen bonding and π–π interactions play a crucial role in molecular recognition. However, the authors did not provide an in-depth discussion of these interactions nor compare them with results from the literature. A more detailed discussion based on relevant studies should be included (e.g., 10.3390/molecules25122841 and 10.3390/molecules28196891).

A: The hydrogen bonding patterns were originally discussed in the manuscript (see the paragraph immediately preceding Figure 5). π–π interactions are impossible for these ligands due to the absence of π systems in their structures.

Reviewer 2 Report

Comments and Suggestions for Authors

In the reviewed manuscript, the authors tackled the challenging task of explaining the phenomena occurring at the lipid-fat interface in relation to the activity of digestive enzymes. The paper attempted to identify a plant extract rich in saponins that was expected to positively influence the activity of these enzymes. These relationships could not be clearly identified, demonstrating the complexity of the topic. The research topic is important both from a cognitive and application perspective. It clearly demonstrates that finding a mixture of natural plant-derived compounds with desirable biological properties is difficult and not always successful.

The paper is clearly written, although the authors did not avoid certain errors and shortcomings, which are listed below:

- L_79-81: the abbreviations RA% and Effect% and CFFA, used for the first time, should be explained in the text.

- Some results are presented in both tables and graphs, which are unnecessary. Authors should choose one format for presenting results.

- Fig. 2, S1, S2, S3, S4; there is no explanation of the letters above the bars.

Author Response

Dear Reviewer,

We greatly appreciate your time and effort in reviewing our manuscript and providing constructive feedback. Below, we provide detailed responses to each of your comments. We hope the revisions have improved the manuscript and made it suitable for publication.

Comments and Suggestions for Authors

In the reviewed manuscript, the authors tackled the challenging task of explaining the phenomena occurring at the lipid-fat interface in relation to the activity of digestive enzymes. The paper attempted to identify a plant extract rich in saponins that was expected to positively influence the activity of these enzymes. These relationships could not be clearly identified, demonstrating the complexity of the topic. The research topic is important both from a cognitive and application perspective. It clearly demonstrates that finding a mixture of natural plant-derived compounds with desirable biological properties is difficult and not always successful.

The paper is clearly written, although the authors did not avoid certain errors and shortcomings, which are listed below:

- L_79-81: the abbreviations RA% and Effect% and CFFA, used for the first time, should be explained in the text.

A: The RA%, Effect%, and CFFA values are too complex to explain briefly in the text. When these values appear in the text for the first time, we added the information that their detailed explanation being provided in the Methods section 5.5.

- Some results are presented in both tables and graphs, which are unnecessary. Authors should choose one format for presenting results.

A: Some results are indeed duplicated. While it would be best to present the values in a graph, the graph could be unclear. Tables and graphs complement each other. The graph shows proportions related to activity differences, while the tables show real values. Please consider retaining the tables and figures.

- Fig. 2, S1, S2, S3, S4; there is no explanation of the letters above the bars.

A: Corrected. There is no statistical significance of differences between bars marked with the same letter. Text was added to the legend of the figures.

Reviewer 3 Report

Comments and Suggestions for Authors

The manuscript is well designed, interdisciplinary, and combines biochemical experiments with computational studies. The topic is relevant and original, with potential applications in digestion and nutrition research. Results are clearly presented, though in some sections the text is overly detailed. The molecular dynamics part requires further clarification and technical refinement.

  • The 100 ns trajectories are informative, but the authors should discuss whether this timescale is sufficient for system equilibration and if replicate runs would strengthen the reliability of the results.
  • Provide clearer information on relaxation steps, number of replicates, and potential limitations of the chosen force field for large glycosides such as escin Ia.
  • The discussion about active site blockade by hederacoside C versus allosteric effects is interesting but partly speculative. It would be helpful to separate direct observations from assumptions and to include additional plots (e.g., distance changes between key residues over time) to support claims.

Author Response

Dear Reviewer,

We greatly appreciate your time and effort in reviewing our manuscript and providing constructive feedback. Below, we provide detailed responses to each of your comments. We hope the revisions have improved the manuscript and made it suitable for publication.

Comments and Suggestions for Authors

The manuscript is well designed, interdisciplinary, and combines biochemical experiments with computational studies. The topic is relevant and original, with potential applications in digestion and nutrition research. Results are clearly presented, though in some sections the text is overly detailed. The molecular dynamics part requires further clarification and technical refinement.

  • The 100 ns trajectories are informative, but the authors should discuss whether this timescale is sufficient for system equilibration and if replicate runs would strengthen the reliability of the results.

A: We agree that longer molecular dynamics simulations for LC complexes could provide additional insights; however, we could not extend them due to time constraints. Our primary objective was to determine whether ligand binding preserves or perturbs the inhibited conformation of the lipase-colipase complex. The simulations clearly showed that binding of hederacoside maintains the inhibited reference structure, whereas escin Ia and cholate induce conformational changes consistent with a shift away from inhibition. We have expanded the discussion in the manuscript (Results and Discussion sections) to clarify this point.

  • Provide clearer information on relaxation steps, number of replicates, and potential limitations of the chosen force field for large glycosides such as escin Ia.

A: Detailed relaxation steps were provided in the Methods section (Computational details) per the Reviewer’s suggestion. A single 100 ns trajectory was generated for each system, which we considered sufficient for comparative purposes given the clear separation in conformational behavior across the different ligands. While replicate runs could further reduce statistical noise, the observed differences, such as stability of LC-hederacoside vs. structural rearrangements of LC-escin Ia/cholate, were robust.  The Reviewer is correct: although the OPLS4e force field has been parametrized for a wide range of organic and biomolecular systems, modeling large glycosidic saponins such as escin Ia remains challenging due to their size and conformational flexibility. Nevertheless, OPLS4e has been widely applied in studies of glycosides (see, for example, https://doi.org/10.1038/s41467-024-53209-1; https://doi.org/10.1039/D4OB01286K; https://doi.org/10.1021/acsomega.5c03915). In response to this comment, we added a statement in the Methods section (Computational details) acknowledging these limitations and clarifying that our results should be interpreted in relative trends rather than absolute quantitative values.

  • The discussion about active site blockade by hederacoside C versus allosteric effects is interesting but partly speculative. It would be helpful to separate direct observations from assumptions and to include additional plots (e.g., distance changes between key residues over time) to support claims.

A: Per the Reviewer’s suggestion, we analyzed the distances between the active site residues. Interestingly, in the cases of escin Ia and cholate, the distance between residues D177 and H264 was significantly reduced compared to the other models. Corresponding plots have been added to Figure 6. In addition, we revised the manuscript text to clearly distinguish speculation from fact by consistently using qualifiers such as “presumably” and “could” where appropriate.

Reviewer 4 Report

Comments and Suggestions for Authors

1. Linking complex extracts to specific saponins requires stronger evidence. Please provide quantitative (or at least semi-quantitative) saponin levels (mg/g extract) for all nine extracts (external standards where possible). Test the pure saponins used in the MD (escin Ia, hederacoside C) in the same lipase assay alongside the extracts to support (or falsify) the MD-derived activation/inhibition narrative. Clearly justify using one compound to represent each chemically complex extract and discuss the limits of that assumption in the Conclusions.
2. Demonstrate time linearity (initial-rate region) and absence of product inhibition over 180 min (such as show time-course curves ± extract).
3. The cholate control (RA% ≈ 1058 ± 30; Effect% ≈ 958 ± 81) is extraordinarily large. Please contextualize relative to physiological bile salt ranges and discuss whether the assay is operating near mass-transfer/surfactant limits.
4. Data tables and figures mostly show “ME (maximal error via total differential method).” The choice of ME is unusual and not standard for biological assays.
5. The Abstract claims antioxidant activity “strongly positively correlated” with total phenols, but no correlation plots/statistics are presented.
6. “10 g … extracted with 80 mL 70% EtOH for 10 days using a reflux condenser” is unclear (continuous reflux for 10 days? maceration under a condenser?).

Author Response

Dear Reviewer,

We greatly appreciate your time and effort in reviewing our manuscript and providing constructive feedback. Below, we provide detailed responses to each of your comments. We hope the revisions have improved the manuscript and made it suitable for publication.

Comments and Suggestions for Authors

  1. Linking complex extracts to specific saponins requires stronger evidence. Please provide quantitative (or at least semi-quantitative) saponin levels (mg/g extract) for all nine extracts (external standards where possible). Test the pure saponins used in the MD (escin Ia, hederacoside C) in the same lipase assay alongside the extracts to support (or falsify) the MD-derived activation/inhibition narrative. Clearly justify using one compound to represent each chemically complex extract and discuss the limits of that assumption in the Conclusions.

A: While we understand the reviewer’s concern, unfortunately, we cannot perform additional experiments, primarily due to the high cost of the standards. Conducting research for these two substances will cost about 20,000 euros. Furthermore, the waiting time to purchase these substances, perform additional experiments, elaborate on results, and prepare manuscripts will delay publication at least a year. Concerning the saponin content in the investigated plants, we can only refer to the previous publication “Saponins as Natural Emulsifiers for Nanoemulsions”  J. Agric. Food Chem. 2022, 70, 6573−6590, where saponin content is given in the most known rich plant sources, for example, Panax ginseng 2−3 (%wt.),  Aesculus hippocastanum 3−6 (%wt.), Hedera helix 5.9 (%wt..). To explain the mechanisms of action of substances that dominate in extracts (our UHPLC qualitative experiments and literature data) and have surface properties, we asked for cooperation from specialists in computer modeling of reactions of biologically active substances.  

  1. Demonstrate time linearity (initial-rate region) and absence of product inhibition over 180 min (such as show time-course curves ± extract).

A: As we pointed out earlier, we are unable to conduct additional experiments due to a lack of time and resources. Preparing extracts, designing the research, and elaborating on results will take several months, up to half a year. Furthermore, the kinetics of lipase action differ from those of other enzymes (proteases) that interact with water-soluble substrates. The kinetics of lipases is so complex that, in my opinion, a simple test cannot demonstrate unambiguous inhibition by products. We appreciate your valuable comment, and we plan such a test in future research, although we realize that this test doesn’t explain many kinetic problems of the reaction. In this work, we will use computational studies to investigate the kinetics of lipolysis and possibly the inhibition of the reaction by the product.

  1. The cholate control (RA% ≈ 1058 ± 30; Effect% ≈ 958 ± 81) is extraordinarily large. Please contextualize relative to physiological bile salt ranges and discuss whether the assay is operating near mass-transfer/surfactant limits.

A: As I mentioned before, the kinetics of action of lipase is very complex and strongly depend on surfactants such as cholic acid. Adding cholate to the reaction mixture increases lipolysis many times, about a tenfold increase in activity. Therefore, RA% and Effect% are high. We develop test conditions based on data from the Internet and (first of all) scientific literature, including data related to the concentration of bile salts and pancreatin in the duodenum during fat digestion. We also performed several preliminary experiments (unpublished data) to investigate the effect of different concentrations of sodium cholate and some extracts on lipolysis.

  1. Data tables and figures mostly show “ME (maximal error via total differential method).” The choice of ME is unusual and not standard for biological assays.

A: The error was calculated using the total differential method due to the transformation of the formulas. The maximal error (ME) well defines the measurement uncertainty.

  1. The Abstract claims antioxidant activity “strongly positively correlated” with total phenols, but no correlation plots/statistics are presented.

A: Corrected. Figure S8 was added, which presents a positive correlation between the general phenol amount and the antioxidant activity.

  1. “10 g … extracted with 80 mL 70% EtOH for 10 days using a reflux condenser” is unclear (continuous reflux for 10 days? maceration under a condenser?).

A: The extraction was carried out at a temperature of 40°C.

Round 2

Reviewer 4 Report

Comments and Suggestions for Authors

accept